# A Multidisciplinary Pathway for the Diagnosis and Prosthodontic Management of a Patient with Medication-Related Osteonecrosis of the Jaw (MRONJ)

Amr S. Bugshan [1],* and Yousif A. Al-Dulaijan [2]

[1] Department of Biomedical Dental Sciences, College of Dentistry, Imam Abdulrahman Bin Faisal University, Dammam P.O. Box 1982, Saudi Arabia

[2] Department of Substitutive Dental Sciences, College of Dentistry, Imam Abdulrahman Bin Faisal University, Dammam P.O. Box 1982, Saudi Arabia

* Correspondence: abugshan@iau.edu.sa

**Abstract:** Background: Medication-Related Osteonecrosis of the Jaw (MRONJ) can occur as an adverse reaction to several antiresorptive medications such as bisphosphonate. It presents clinically as a necrotic exposed bone. Several factors including tooth extraction and ill-fitting dentures increase the risk of osteonecrosis development. Case Report: A 72-year-old female who had an ill-fitting partial denture that caused an exposed necrotic bone and traumatic ulcer on the left posterior mandible. Bony sequestrums were removed and submitted for histological examination, which confirmed the diagnosis of MRONJ. Conclusions: This case illustrates the importance of identifying all risk factors associated with MRONJ by dentists to reduce its development in patients receiving antiresorptive medications. Moreover, patients at risk of MRONJ development should be screened carefully on a regular basis and all dental risk factors should be adjusted or removed.

**Keywords:** MRONJ; bisphosphonate; sequestrum; ill-fitting dentures

## 1. Introduction

Several medications with antiresorptive qualities are widely used to control and prevent bone loss in patients with osteoporosis and osteopenia to reduce the fracture risk and skeletal-related problems such as pathological fractures and pain associated with bone metastatic malignancies [1,2]. These medications include antiresorptive and antiangiogenic medications. Antiresorptive medications include bisphosphonates (BPs) and RANK ligand inhibitors (denosumab) [3].

Osteonecrosis of the jaw as an adverse reaction to bisphosphonate (BRONJ) was first reported in 2003 and 2004 [4]. Due to the growing number of jaw osteonecrosis cases associated with other antiresorptive (RANK ligand inhibitors) and antiangiogenic medications, the American Association of Oral and Maxillofacial Surgeons (AAOMS) wants to replace BRONJ with medication-related osteonecrosis of the jaw (MRONJ) [3]. MRONJ is defined as a non-healing exposed bone that has persisted for more than 8 weeks in patients who have or are currently taking antiresorptive or antiangiogenic medications with no history of radiation therapy in the head and neck region [3,5].

BPs are grouped into two categories based on the route of administration: Oral BPs—mainly used in the treatment of metabolic bone disorders including osteoporosis and osteopenia, and intravenous (IV) BPs—that are therapeutically employed for the treatment of metastatic bone-related malignancies [1]. BPs reduce the capability of osteoclasts to resorb bone at low concentrations, while high concentrations of BPs enhance the apoptotic cell death of the osteoclasts [6].

The frequency of MRONJ in cancer patients receiving IV BP is 100 times higher than those taking oral BPs for osteoporosis and the prevalence increases over the exposure

time in both oral and IV BPs [4]. The prevalence of MRONJ in patients taking oral BP increases drastically after four or more years of BP exposure [7]. The exact mechanism of MORNJ development is not yet thoroughly understood, however, several hypotheses on the pathophysiology of MORNJ have been suggested. These include anti-angiogenesis, infection, bone remodeling suppression, and genetic susceptibility [8].

Generally, dentists should be able to identify all the risks associated with the dental treatment of patients receiving antiresorptive or antiangiogenic medications. Tooth extraction is considered one of the major risk factors for Osteonecrosis of the Jaw (ONJ) development [9]. Moreover, other risk factors such as pre-existing periodontal inflammation or periapical pathosis [10], systemic conditions such as anemia and diabetes [11], corticosteroid intake [11], and tobacco [12] have been reported as risk factors for MRONJ.

Ill-fitting dentures are reported to be associated with an increased risk for MRONJ development in patients receiving BP [12]. Mucosal barrier injury can result from a constant trauma caused by ill-fitting dentures permitting various pathogens to enter the bone and, as a result, increasing the risk for MRONJ [12,13]. Therefore, dentures should be carefully examined and evaluated for traumatized areas in patients at risk of MRONJ development. Moreover, the fabrication of a new removable prosthesis or relining the old ill-fitting dentures should be considered to minimize trauma and reduce the risk of MRONJ [13].

In this paper, we report on a case of MRONJ in an osteoporotic patient with long-term use of oral BP, emphasizing the radiographic and histopathological findings and modifying the prosthodontic approach to reduce the risk of MRONJ development.

## 2. Case Report

### 2.1. History and Clinical Examination

A 72-year-old female complained about a recurring infection and a painful non-healing exposed bone on the left buccal shelf of the posterior edentulous mandibular alveolar ridge. Her symptoms initially started a year before as a 1-mm exposed bone on the buccal shelf of the posterior mandibular alveolar ridge with swelling on the vestibule and pus discharge. She was treated with antibiotics. The onset of her symptoms started after fabrication of mandibular Removable Partial Denture (RPD) as opposed to the maxillary Completed Denture (CD), which was not probably fitted especially on the left mandibular side. The patient's medical history consisted of osteoporosis, hypertension, hypercholesterolemia, controlled type 1 diabetes, history of hepatitis C, and acid reflux. Her medications included valsartan/hydrochlorothiazide (160/12.5 mg), atorvastatin (20 mg), dapagliflozin (10 mg), insulin lispro injection (Humalog), pantoprazole (40 mg). Furthermore, the patient was treated with oral BP, Ibandronate 150 mg for more than 20 years (Boniva, Roche Pharmaceuticals, Basel, Switzerland), and then replaced by Teriparatide (Forteo, Eli Lilly and Co., Indianapolis, IN, USA) 20 mcg subcutaneous injection once a day. Extraoral findings were unremarkable. Intraoral examination revealed 15 × 6 mm exposed bone on the left posterior alveolar ridge of the mandible with an 8 × 7 mm traumatic ulcer on the lingual side (Figure 1). Furthermore, edentulous alveolar ridge in the maxilla and bilateral posterior molars, bilateral central incisors, and left lateral incisor are missing in the mandible (Kennedy's class II modification I) were observed. All remaining teeth are periodontally involved with severe occlusal surface attrition.

### 2.2. Radiographic Findings

Orthopantomogram (OPG) examination showed an ill-defined radiolucent lesion in the molars area of the left mandible (Figure 2A). A multi-detector computed tomography (CT) scan performed with bone window showed an area with cortical bone disruption and mottled radiolucency of the left posterior mandible (Figure 2B,C).

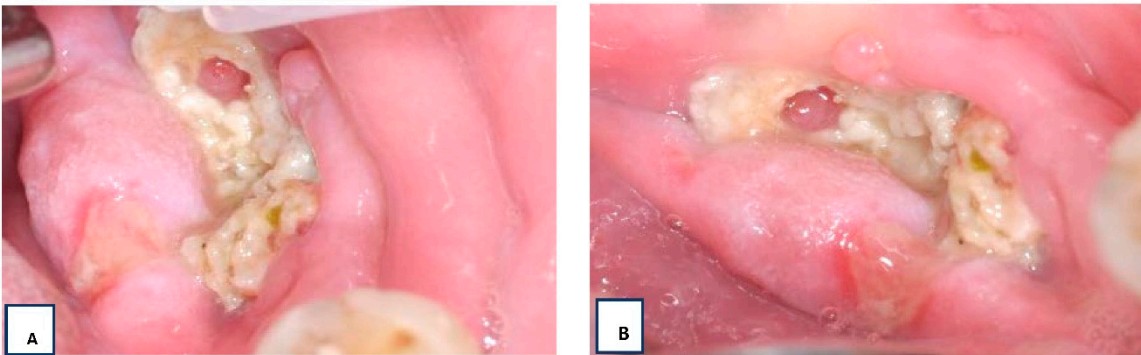

**Figure 1.** Medication (bisphosphonate)-Related Osteonecrosis of the Jaw (MRONJ). (**A**,**B**) A necrotic exposed bone on the left posterior alveolar ridge of the mandible with traumatic ulcer on the lingual side.

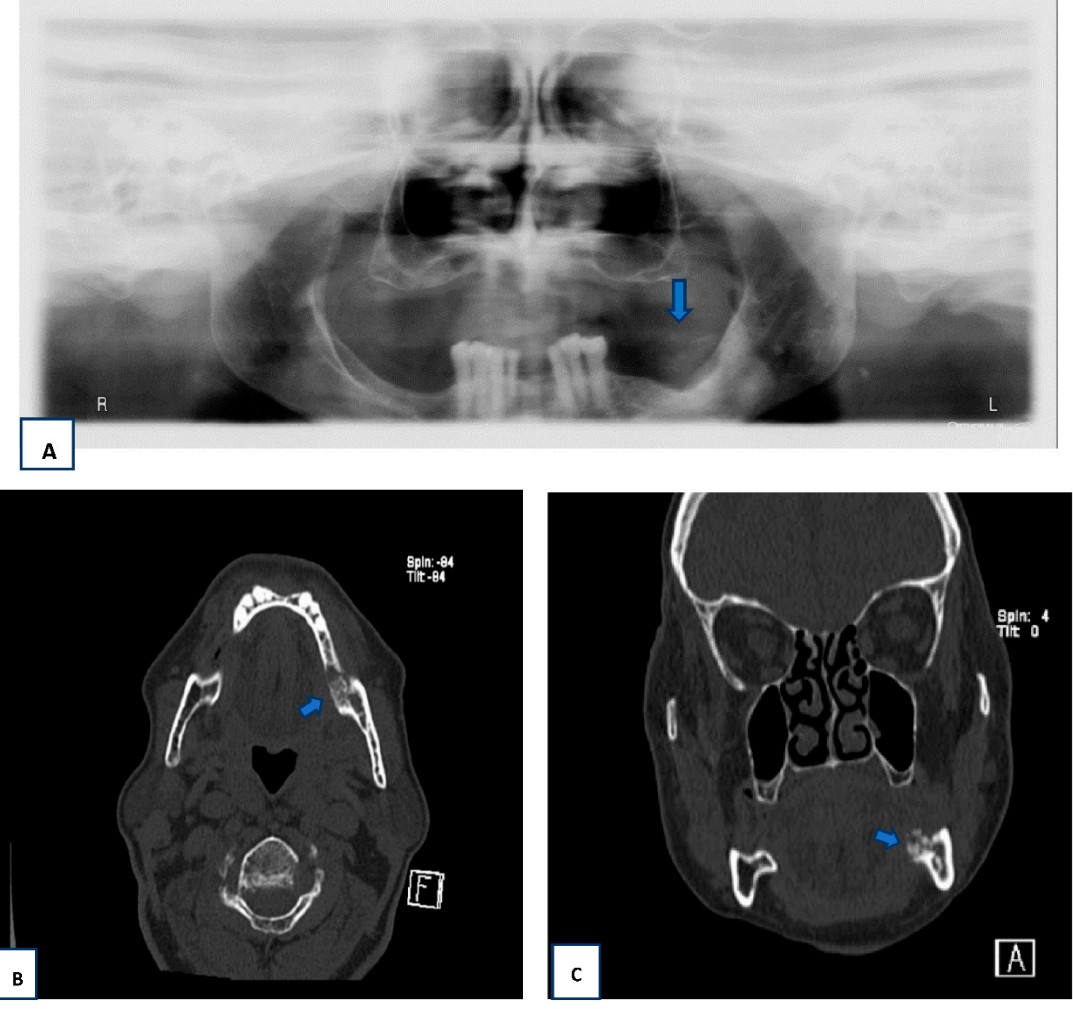

**Figure 2.** (**A**) OPG radiograph showing necrotic bone hyperplasia on the left posterior alveolar ridge of the mandible (marked by blue arrow). Multidetector CT scan bone window showing the lytic bone lesion (marked by blue arrow) (**B**) Axial slice of the level mandible view and (**C**) Coronal view at the level of orbit and maxillary sinuses. R: right, L: left, F: frontal, and A: Axial.

*2.3. Histopathological Findings*

Fragments of necrotic bony (sequestra) were removed and fixed in 10% formalin for histopathological analysis (Figure 3A). Microscopic examinations revealed fragments of

non-viable lamellar bone rimmed by basophilic bacterial biofilm (Figure 3B). Dense colonies of bacteria surrounded by granulation tissue with abscess (Figure 3C).

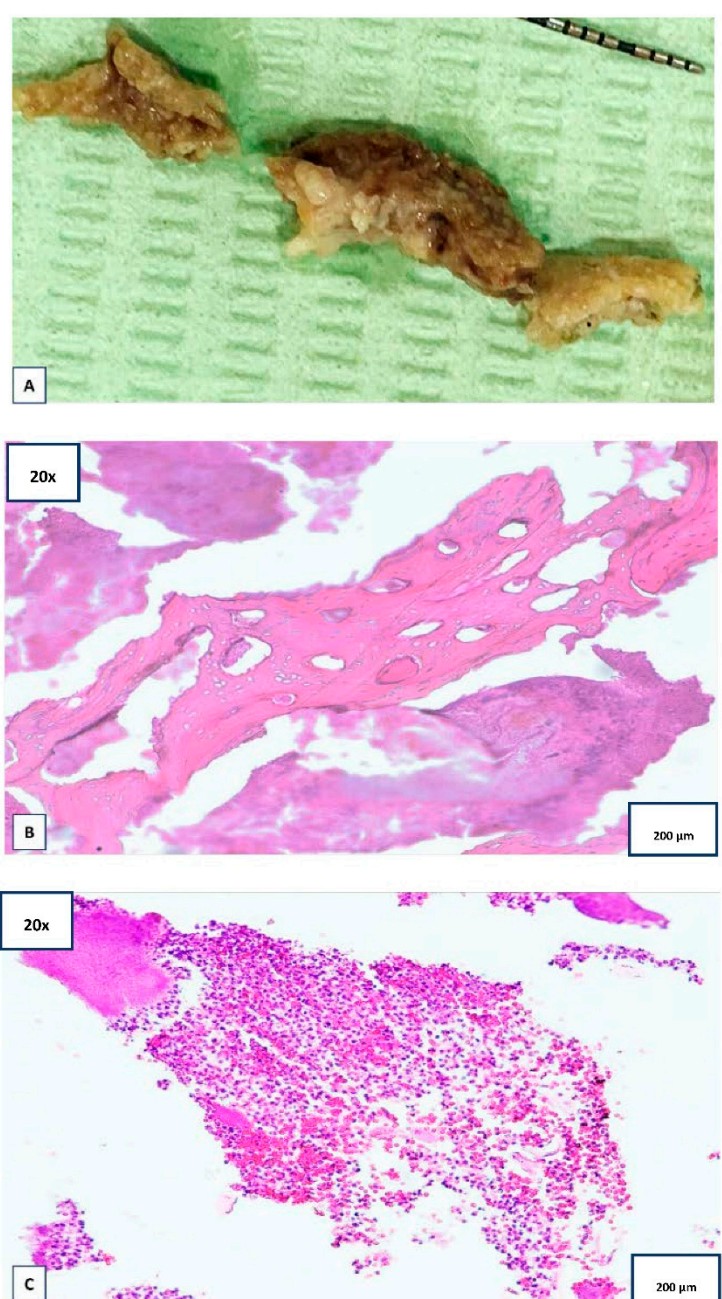

**Figure 3.** (**A**) Necrotic bony fragments removed from the left posterior alveolar ridge of the mandible. (**B**) A photomicrograph of a hematoxylin and eosin-stained section showing bony sequestrum surrounded by dense colonies of bacteria (20×). (**C**) A photomicrograph of a hematoxylin and eosin-stained section showing bacterial colonies surrounded by granulation tissue and abscess (20×).

### 2.4. Diagnosis

The patient had the exposed bone for the duration of 8 weeks in the mandibular region without a history of a previous radiotherapy. Therefore, the clinical, radiographic, and microscopic findings confirmed the diagnosis of MRONJ resulting from long-term bisphosphonate therapy for osteoporosis.

*2.5. Management*

According to AAOMS, the patient is classified as stage 2 (Table 1) [3].

**Table 1.** Stages of MRONJ based on the disease progression.

| Stages | Features |
|---|---|
| **At risk** | No exposed necrotic bone clinically. patients have been treated with MRONJ-related medications. |
| **Stage 0** | Non-specific clinical findings, radiographic changes, and symptoms without clinical evidence of bony exposure. |
| **Stage 1** | Clinical evidence of asymptomatic exposed necrotic bone without signs of infection. |
| **Stage 2** | Clinical evidence of symptomatic exposed necrotic bone with signs of infection i.e., pain, erythema, or purulent discharge. |
| **Stage 3** | Clinical evidence of symptomatic exposed necrotic bone with signs of infection that extend beyond the region of alveolar bone. |

This case was initially managed by a conservative surgical approach, bone debridement, and gentle removal of necrotic bony fragments (sequestrectomy) along with the prescription of an antibiotic (amoxicillin 500 mg twice/day for 7 days), analgesic, and an antibacterial (0.12% chlorhexidine) mouth rinse. The patient was also instructed to lightly irrigate the lesion with a 5 ML disposable syringe filled with normal Saline (Sodium Chloride 0.9%) and discontinue the use of her existing mandibular RPD. After the area was fully healed with no evidence of exposed necrotic bone radiographically (Figure 4) and clinically (Figure 5A,B), the patient was referred to a prosthodontist to fabricate a new well-fitting removable denture with careful evaluation of traumatic areas to reduce the risk of mucosal irritation, ulceration, and possible development of MRONJ again.

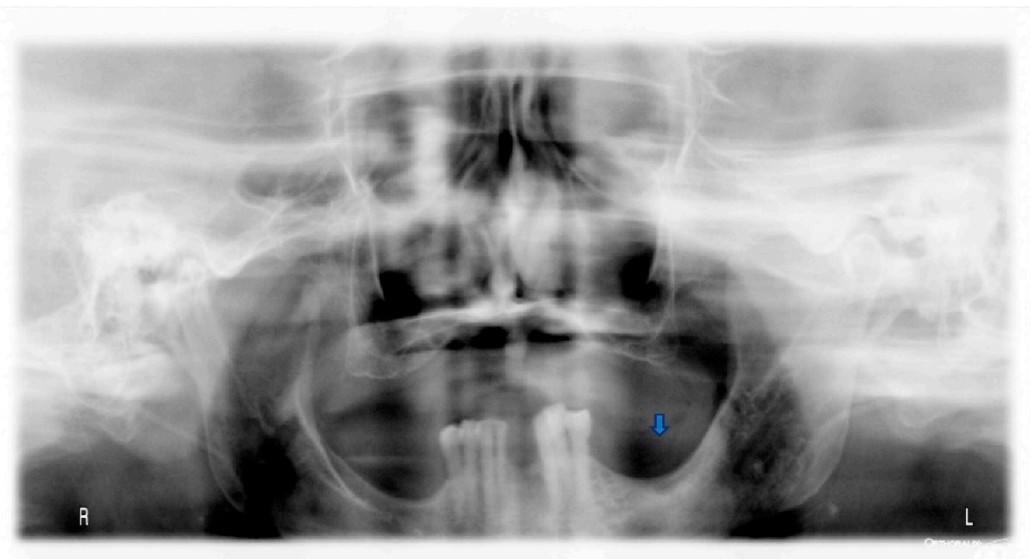

**Figure 4.** OPG radiograph showing complete healing with no evidence of necrotic bone on the left posterior alveolar ridge of the mandible as marked by blue arrow. R: right and L: left.

During the initial prosthodontics consultation, the existing mandibular RPD was evaluated and the possible cause of trauma to the underlying tissue was indicated as exposed metal framework mesh and a lack of sufficient acrylic resin underneath it (Figure 6A).

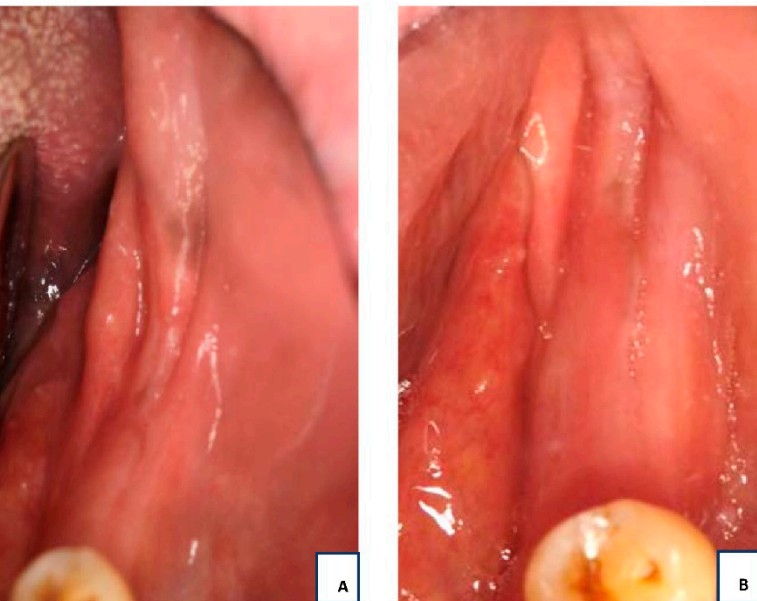

**Figure 5.** Clinical photos showing complete healing with no evidence of necrotic bone on the left posterior alveolar ridge of the mandible. (**A**) Follow up after management and before dentures fabrication. (**B**) 3 months post dentures insertion.

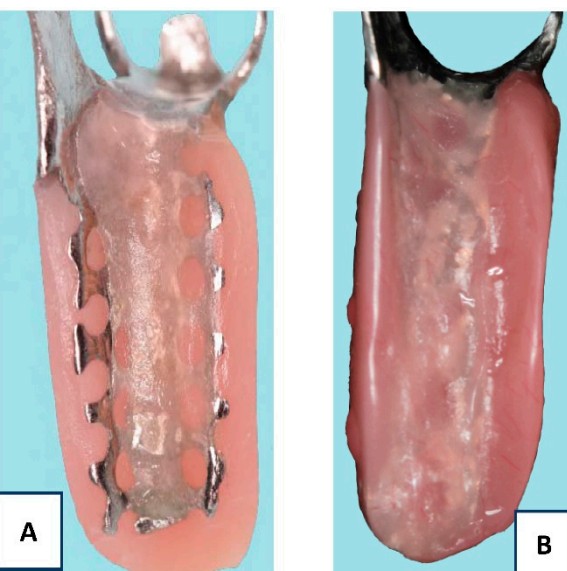

**Figure 6.** Mandibular RPD denture bases. (**A**) Old RPD with metal framework exposure. (**B**) Newly fabricated RPD with proper denture base extension and acrylic thickness.

The patient was advised to discontinue the use of RPD to allow tissue healing. After 6 months, the new dentures were fabricated according to standard protocol in which the maxillary and mandibular primary impressions were made by alginate. Then, light-cured resin material was used to fabricate custom trays. Following that, the maxillary final impression for CD and Mandibular final impression for RPD were made with polyether impression material (Impregum™, 3M ESPE, Dental Products, St. Paul, MN, USA) after performing the border molding with the impression compound modeling plastic (Impression Compound Green Sticks; Kerr Corp) without any overextension. To fabricate the metal framework for mandibular, a relief space beneath the minor connectors and underlying tissue was established by placing a sheet of relief wax on the original master cast and duplicating it in the refractory cast. Following that—in a separate visit—metal framework of the mandibular RPD was tried in the mouth and the vertical dimension of occlusion

and centric occlusion were recorded. In another appointment, the wax maxillary CD and mandibular RPD dentures were tried on. The extension of denture bases was evaluated and the occlusion was checked. Before insertion, laboratory re-mounting was performed to reduce the processing errors. Following that, the intaglio surface of dentures was assessed and any rough objections were removed. The proper extension of the mandibular RPD denture base was evaluated (Figure 6B). Following insertion, the patient was encouraged to seek regular follow-up appointments. During each of these appointments, oral hygiene instructions were re-emphasized, the underlying tissues were examined, and the area of high pressure was marked by using silicone paste (pressure indicated paste, Keystone Industries, Cherry Hill, NJ, USA) then adjusted by acrylic bur. At 3-month follow-up, there was no evidence of exposed necrotic bone clinically (Figure 5B).

## 3. Discussion

BPs bind to the hydroxyapatite and deposit in the bone. During the bone remodeling cycle, the BPs are released from the hydroxyapatite and inhibit osteoclast-mediated bone resorption [14]. Two theories are proposed for MRONJ pathogenesis. In the first theory, the MRONJ originates from the bone through osteoclasts inhibition and then spreads to the soft tissue (inside-outside). The other less likely theory, MRONJ develops primarily from soft tissue necrosis and then spreads to the bone [15].

In a cohort study, Lo et al. found the ONJ prevalence was 0.10% among the patients receiving chronic oral BPs. The prevalence was higher among patients receiving oral BPs after 4 years or more (0.21%) compared with those with 4 years of exposure (0.21% vs. 0.04%, respectively) [6]. Another study conducted by Bamias et al. reported that the incidence of ONJ increased from 1.5% among patients receiving BP for 4 to 12 months to 7.7% for those who were treated for 37 to 48 months [16].

The pathogenesis of MRONJ is multifactorial, however, the presence of a local factor plays an important role in the initiation of osteonecrosis [17]. In a study by Yazdi and Schiodt, the two main factors associated with MRONJ were dental extraction and dental prosthesis [18]. In another case-controlled study, the risk factors for ONJ development were evaluated in 20 breast cancer patients receiving BP. They found that 8 of the ONJ cases (40%) were wearing dentures ($p = 0.064$) [12]. In a prospective study by Bamias et al., 2 of 17 patients diagnosed with ONJ in the mandible were wearing dentures, and in both cases, ONJ developed with multiple myeloma after receiving BPs [16].

In this case report, the radiographs were used to confirm the diagnosis and evaluate the extent of the lesion besides the clinical examination. Thus, several authors have emphasized the importance of the radiographic features of MORNJ and suggested that the radiographical findings should be added to the AAOMS's staging [3,19,20].

During the comprehensive oral examination, the treating dentist or prosthodontist should be able to identify all possible risk factors for MORNJ such as bony spikes, spicules, and exostoses, and pre-prosthetic surgeries that may be recommended for some patients if needed [21]. For denture fabrication, extra attention should be paid to denture relief areas, proper denture base extension, and impression techniques, and occlusion to minimize the trauma to underlying tissue during mastication [21]. Additionally, efforts should be made by both the dentist and patient regarding the development of good oral hygiene protocol.

To reduce the risk of MRONJ, all patients must be educated about the risk of MRONJ and the importance of maintaining good oral hygiene. They also must be carefully examined prior to the administration of antiresorptive therapy. Comprehensive dental treatment should be performed to reduce the need for dental procedures during the treatment. Moreover, all invasive dental treatments such as extraction of non-restorable teeth and necessary periodontal surgeries should be performed. Also, the existing dentures should be carefully examined, and poorly fitting dentures must be adjusted or replaced to reduce tissue trauma. For patients who already have started the treatment, invasive dentoalveolar surgical procedures should be avoided as much as possible to reduce the risk of MRONJ. Therefore, endodontic treatment is preferable for non-restorable teeth instead of extraction [3,22].

## 4. Conclusions

In this case, gentle surgical removal of the loose sequestra, together with chlorohexidine mouthwash and systemic antibiotics were necessary to achieve complete healing of the exposed bony area. Removable prostheses have been used as a predictable treatment option in the management of this patient. However, caution should be exercised for those at risk of developing MRONJ through careful and periodic examinations and reducing constant tissue trauma by fabrication of well-fitting dentures.

**Author Contributions:** Conceptualization, A.S.B. and Y.A.A.-D.; formal analysis, A.S.B. and Y.A.A.-D.; writing—original draft preparation, A.S.B. and Y.A.A.-D.; writing—review and editing, A.S.B. and Y.A.A.-D.; supervision, A.S.B. and Y.A.A.-D. All authors have read and agreed to the published version of the manuscript.

**Funding:** This research received no external funding.

**Institutional Review Board Statement:** Not applicable.

**Informed Consent Statement:** Informed consent was obtained from the patient involved in the study.

**Conflicts of Interest:** The authors declare no conflict of interest.

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
