# Peer review of "A Multidisciplinary Pathway for the Diagnosis and Prosthodontic Management of a Patient with Medication-Related Osteonecrosis of the Jaw (MRONJ)"

_applsci, doi:10.3390/app12168202_

Round 1

Reviewer 1 Report

First of all, thank you for the opportunity to review this manuscript.

The aim of the following case study was to present Medication-Related Osteonecrosis of the Jaw in 72-year-old female.

The following are suggestions for the present manuscript:

Authors stated: „She was treated with antibiotics.“ - Which antibiotics, and for how long?

Check that the references are written in accordance with the journal's instructions.

Reviewer 2 Report

This clinical case does not add anything new to what is known about the disease. It is not innovative and, as such, I understand that it should not be published.

The translation must be reviewed by a native English language expert.

Title

The title must not have acronyms

Case Report

What type of diabetes? Are you under control?

What medications does the patient take?

Which teeth are missing?

What other intraoral diseases are found? caries? Periodontitis?

“…Table 1: Stages of MRONJ based on the disease progress…” - Table 1 must be on top of the table with the title and legend below.

Which antibiotic was prescribed? For how long?

Figure 4: the orthopantomography must be a single figure, It must not be with the figures that correspond to the prostheses. Authors must make a legend

Figure 5: remove: “…Photos of the…”

Conclusions

I consider the conclusions to be incomplete considering the intervention performed as a whole - should the lesion be removed?

References

The authors present two references 1? I didn't understand...

Reviewer 3 Report

Dear authors, I was pleased to review your case report regarding mjorns. I found it interesting. However, some aspects need to be clarified to increase the quality of your article.

1) Figure 1 A appears blurred. The authors should increase the definition of the photo.

2) In figure 2 C the arrow has been placed in the wrong position. 

3) What surgical protocol was used to remove the bone fragment? The authors should describe it in the case report.

4) After removal of the bone segment, how was it preserved before histological analysis?

5) In the histological analysis performed, what type of dye was used?

6) In microscope images, the magnification must always be stated. Please add this information to the images and the image caption.

7) The micron scale of the histological photos is not clearly visible. Please enlarge the text.

8) During the healing period, was the patient prescribed antibiotic therapy or application of a gel to speed up healing?

9) The authors state that during wound healing the patient was told to wear rpd discontinuously. Was the prosthesis, which had an exposed metal frame, relined before being returned to the patient? How was it relined? With what material?

10) In the introduction, the authors should briefly describe the mechanism of action of bisphosphonates.

Reviewer 4 Report

Authors represent the case study of a 72 years old female patient who had an ill-fitting 12

partial denture it caused an exposed necrotic bone and traumatic ulcer on the posterior mandible, histological analysis confirms MRONJ (medication related osteonecrosis of the jaw). 

The paper is worth publishing because it shows the importance of identifying all risk factors associated with MROJ. The dentists should be able to identify all the risks connected with dental treatment of patients receiving antiresorptive or antiangiogenic medications.

The Introduction is well written, but one recent reference is missing 

Mampei Kawahara, Shinichiro Kuroshima and Takashi Sawase.  Clinical considerations for medication related osteonecrosis of the jaw: a comprehensive literature review, International Journal of Implant Dentistry (2021) 7:47. https://doi.org/10.1186/s40729-021-00323-0

The methods, radiographic OPG, CT and histopathology are selected correctly for the diagnosis MROJ confirmation. Extensive discussion leads to the conclusion of the case 

Study that gentle surgical removal of the loose sequestra, together with chlorohexidine mouthwash and systemic antibiotics were necessary to achieve complete healing of the exposed bony area.

Round 2

Reviewer 2 Report

This clinical case does not add anything new to what is known about the disease. It is not innovative and, as such, I understand that it should not be published.
